# Quality of Life in Oral Cancer Patients in Greek Clinical Practice: A Cohort Study

**DOI:** 10.3390/jcm11237235

**Published:** 2022-12-06

**Authors:** Maria Lavdaniti, Ioannis Tilaveridis, Dimitra Palitzika, Athanasios Kyrgidis, Stefanos Triaridis, Konstantinos Vachtsevanos, Angeliki Kosintzi, Konstantinos Antoniades

**Affiliations:** 1Nursing Department, International Hellenic University, 57400 Thessaloniki, Greece; 2Oral and Maxillofacial Surgery, Aristotle University of Thessaloniki, 54124 Thessaloniki, Greece; 3Nursing Director, Papanikolaou Hospital, 57010 Thessaloniki, Greece; 4Department of Oral and Maxillofacial Surgery, School of Medicine, Faculty of Health Sciences, Aristotle University, 54124 Thessaloniki, Greece; 51st Department of Otorhinolaryngology, Aristotle University of Thessaloniki, 54124 Thessaloniki, Greece; 6Oral and Maxillofacial Surgery, Papanikolaou Hospital, Aristotle University of Thessaloniki, 57010 Thessaloniki, Greece; 7Special Education, Aristotle University of Thessaloniki, 54124 Thessaloniki, Greece

**Keywords:** quality of life, oral cancer, Greece

## Abstract

Cancer of the oral cavity is one of the most common cancers all over the world. Oral cancer and its treatment impacts on patients’ Quality of Life (QOL). The purpose of the present study was to assess oral cancer patients’ QOL after the completion of surgical therapy, and to investigate factors affecting it. This was a prospective cohort study, conducted at the Department of Oral & Maxillofacial Surgery, of a large general public hospital in Northern Greece. The sample consisted of 135 consecutive eligible cancer patients. Three distinct questionnaires were used. The first one included questions regarding the participants’ demographic characteristics and relevant clinical information. The second comprised the European Organization for Research and Treatment core module (EORTC QLQ-C30) and its head and neck module EORTC QLQ-H&N35. The third was the Functional Assessment of Cancer Therapy–General (FACT-G) assessment of quality of life. We also included the physician-completed Karnofsky scale to assess the functional status of the participants. We found that location of the tumor affects QOL and specifically social contact (H = 17.89, *p* = 0.001), on the first assessment, and nutritional supplements (H = 22.49, *p* = 0.000), on the fourth assessment. QOL in patients deteriorates immediately after treatment but significantly improves over time. Health care professionals should take into account these results and arrange care plans in order to find ways to increase patients’ QOL.

## 1. Introduction

Cancer of the oral cavity is one of the most common cancers all over the world both in developing countries and in the developed world [1]. In Greece, the estimated annual incidence of OC is 600–650 new cases [2].The treatment modalities of oral cancer include surgery, chemotherapy, radiotherapy (RT), or a combination of them [3], depending on the stage of the cancer as determined by clinical diagnosis [4].

According to the World Health Organization QOL Group, quality of life (QOL) is defined as “an individual’s perception of his/her position in life in the context of the culture and value systems in which he/she lives and in relation to his/her goals, expectations, standards, and concerns”. This definition considers that QOL is a subjective concept and depends on an individual’s judgment [5].

Oral cancer and its treatment impacts patients’ QOL [6]. QOL in oral cancer patients has received increasing attention due to a rising number of new cases and an improved survival rate [7].

Several studies have examined the quality of life in oral cancer patients. One of them examined quality of life in oral cancer patients who underwent reconstruction in the oral cavity after six months [8]. Another study compared the quality of life in elderly and non –elderly oral cancer patients pre-treatment, at treatment completion, and at 1, 3, and 6 months after the surgery [9]. They concluded that type of reconstruction and age influence the improvement of quality of life. Further recent studies were focused on assessing QOL in oral cancer patients and their symptoms from cancer treatment [4,7]. Wang et al. (2020) who used EORTC-QLQ C30 and H&N 35 questionnaires found that oral cancer patients self-reported many adverse effects such as swallowing, dry mouth and sticky saliva. They found that cancer stage, disease duration, duration since therapy, treatment type, and the frequency of surgery, radiation and chemotherapy affected QOL [4]. In another cross-sectional study that was conducted in 97 patients, the most reported symptoms were xerostomia, pain and dysphagia. QOL had been reported to deteriorate immediately after treatment and to improve over time [7].

Undoubtedly, there is a growing interest in the QOL of oral cancer patients after their treatment [8,9]. In Greece, to the best of our knowledge, there are a limited number of studies assessing QOL in oral cancer patients [10,11], which is what stimulated our interest to further investigate this construct. The purpose of the present study was to assess oral cancer patients’ QOL after the completion of surgical therapy, and to investigate factors affecting it. Specifically, we aimed to assess the following research questions:Does the level of QOL change after the completion of surgery therapy?Do different sociodemographic and clinical factors affect QOL in oral cancer patients?

### 1.1. Study Design and Sample 

This was a prospective cohort study conducted in the Department of Oral & Maxillofacial Surgery, of a large general public hospital in Northern Greece between October 2016 and November 2019. The cohort consisted of 135 consecutive eligible head and neck cancer patients who were operated. Sample size calculation in order to avoid a type I or a type II error revealed the need for 130 cancer patients. We assume that the greater the proportion of the whole population that is studied, the closer we will get to the true answer for that population [12]. The eligibility criteria were: age over 18 years, histologically documented diagnosis of cancer, therapeutic operation at baseline, planned and completed adjuvant radiotherapy, willingness to participate in the study, mental ability to complete the questionnaire, and ability to speak and write in the Greek language and no previous neurological and/or psychiatric history.

Patients completed the questionnaire at four timepoints: prior to surgery baseline as well as three, six, and 12 months following surgery. All eligible patients provided written, informed consent before completing a structured questionnaire. Patient and treatment characteristics were collected from hospital records. The Ethical Committee approved the study (73/15-1-2020).

### 1.2. Instruments

Three distinct questionnaires were used. The first one included questions regarding the participants’ demographic characteristics and relevant clinical information. The second comprised of the European Organisation for Research and Treatment core module (EORTC QLQ-C30) and its head and neck module EORTC QLQ-H&N35. The third was the Functional Assessment of Cancer Therapy–General (FACT-G) assessment of quality of life. We also included the physician-completed Karnofsky scale to assess the functional status of the participants.

The EORTC QLQ-C30 consists of five functional dimensions on physical, role, cognitive, emotional, and social functioning, three symptom dimensions (fatigue, pain, and nausea and vomiting), global health status (QL), and single items assessing additional symptoms commonly reported by cancer patients, together with the perceived financial impact of the disease. All items were answered on a 4-point Likert scale ranging from ‘not at all’ to ‘very much’. The questions regarding global QOL were scored on a 7-point scale. Higher scores for the functional scales and global QOL represent a higher level of functioning and high quality of life, respectively. Higher scores for the symptoms represent a greater extent of symptoms [13,14]. Overall internal consistency was acceptable to excellent with a Cronbach’s α of 0.7088 to 0.852.

To provide more detailed information on specific clinical populations, the EORTC QLQ-C30 can be used in conjunction with supplementary questionnaire modules. We used the EORTC QLQ-H&N35, a 35-item questionnaire, for the current study’s head and neck-specific module [15]. Seven multiple-item symptom scales (pain, swallowing, taste/smell, speech, social eating, social contacts, and sexuality) and six symptom items (teeth problems, trismus, dry mouth, sticky saliva, cough, and feeling ill) are generated using this questionnaire [15]. The first 30 items are scored on a four-point Likert scale (‘not at all’, ‘a little’, ‘quite a bit’, and ‘very much’), while the final five items are scored on a no/yes basis. Item scores were linearly transformed to a 1 to 100 scale. Higher scores for the functional scales and global QOL represent a higher level of functioning and high quality of life, respectively. Higher scores for the symptoms represent a greater extent of symptoms [13,14]. Cronbach’s alpha for this study was 0.88. The module addresses the time period ‘during the last week’. The EORTC QLQ-C30 and QLQ-H&N35 have both been tested and validated for the Greek population and have been found to be sufficiently valid and reliable [15,16]. The Spearman–Brown coefficient was 0.827 and 0.856 for QLQ-C30 and HN35, respectively.

FACT-G consists of 27 items that measure the four dimensions of quality of life: physical well-being, social/family well-being, emotional well-being, and functional well-being. Each question of the scale uses a five-point scale (0 = not at all to 4 = very much). The overall scores of all items in the subscales ranged from 0–108, with higher scores indicating better quality of life [17]. In the present study, for FACT-G., the Cronbach’s alpha ranged between 0.71 to 0.80 for each subscale. The Spearman–Brown coefficient was 0.843. It has been translated into the Greek language (https://www.facit.org/measure-languages/FACT-G-Languages, accessed on 15 January 2016).

### 1.3. Psychometric Evaluation of the Instruments

Internal consistency was examined with Cronbach α, a measure for the mean correlation between all items belonging to a common construct. Test-retest reliability refers to the stability of a score derived from serial administrations of a measure by the same rater. Repeated measurements are made by the same individuals, presumably with a time interval long enough to ensure independence. In the present study, we used the Spearman–Brown split-half reliability coefficient. 

Descriptive statistics were used for demographic characteristics. Kolmogorov–Smirnov and Shapiro–Wilks tests were conducted and Q-Q plots were constructed to investigate normality in data. Skewness and kurtosis were also examined. Data distribution did not meet the criteria for normality in most of the variables, so non-parametric tests were used. In order to explore gender differences, the Mann–Whitney test was used. The Kruskal–Wallis test was used to explore differences regarding demographic variables. Linear regression analysis was chosen to check for the effects of clinical variables on performance status, after testing for homogeneity by using Levene’s test. 

In this cohort study of independent cases, assuming a 1:1 ratio among different outcomes, a failure rate among patients with the exposure of 0.2, and a true measured failure rate of 0.5, we will need to study 38 times 2 = 76 subjects in order to reject the null hypothesis that the failure rates among different groups are equal, with probability (power) 0.8. With regard to paired measurements, assuming continuous variables with standard deviation 0.4 of the mean and a true difference in the mean response of matched pairs of 0.3, we will need to study 16 pairs of subjects to be able to reject the null hypothesis that this response difference is zero, with probability (power) 0.8. 

The Type I error probability associated with these tests of this null hypothesis is 0.05. 

The statistical package for social sciences statistical software (version 25.0, IBM SPSS Statistics for Windows, Armonk, NY, USA: IBM Corp) was used for data analysis.

## 2. Results

Demographic and clinical characteristics are displayed in Table 1 and Table 2, respectively. The mean age of our sample was 65.5 (±13.4) years, with a range from 20 to 91 years of age. With regard to demographic characteristics, participants were typically male (n = 84, 62.2%), married or with partner (n = 63, 47.8%) and in retirement (n = 79, 59.8%). Most patients had undergone selective lymphadenectomy (77.8%), whereas 20% had received modified radical neck dissection (MRND) I-III. With regard to location and stage, tongue was the most common site affected (41.5%), while most of the patients were initially diagnosed with stage IV of cancer (52.7%). 

Internal consistency of the EORTC QLQ-C30 scale found to be adequate. The Cronbach α for the 9 items was 0.865. Test-retest reliability was also found to be adequate. The Spearman–Brown split-half reliability coefficient was 0.869.

The Mann–Whitney test revealed that there were not statistically significant differences between gender, educational status and QOL at all timepoints. Age positively correlated with emotional functioning on the EORTC QLQ-C30 scale (Rho = 0.280, *p* = 0.001), on the first assessment.

Regarding the clinical characteristics, Kruskal–Wallis was used to investigate the differences between the location of tumor (tongue, floor, jaw, cheek mucosa, lip), tumor stage, tumor differentiation grade G and QOL (all scales) at all time points. There were statistically significant differences found between location of tumor and questions “HN-Trouble with social contact” (H = 17.89, *p* = 0.001), on the first assessment, and “HN-Nutritional supplements” (H = 22.49, *p* = 0.000), on the fourth assessment (QLQ-HN35). Additional statistical analysis revealed that patients with cancer of the lip had worse social contact (29.7 ± 24.4) than the patients with cancer of the tongue (7.8 ± 14.8) and than the patients with cancer of the jaw (4.8 ± 9.7). In addition, patients with cancer of floor of the mouth frequently used nutritional supplements (26.7 ± 14.1) as compared to patients with tongue cancer (4.3 ± 11.4). 

Stage of the tumor positively correlated with the symptoms of fatigue (r = 0.400, *p* = 0.000) and pain (r = 0.431, *p* = 0.000), as indicated by responses to the EORTCQLQ-30 questionnaire on the third assessment. Furthermore, on the third assessment, stage positively correlated with the subscale “General functioning” of the FACT-G questionnaire (r = −0.377, *p* = 0.001). There is no statistically significant difference in different time points regarding the tumor differentiation grade G. The Mann–Whitney test revealed that there was a statistically significant difference between patients who underwent lymphadenectomy and the subscale of cognitive functioning of the questionnaire EORTCQLQ-30 (U = 493.0, *p* = 0.001) at the third measurement point (Table 3).

Across-subjects analysis for all patients indicated that a statistically significant difference existed in most of the EORTC- QLQ 30 subscales scores (global health status, physical functioning, role functioning, emotional functioning, cognitive functioning, social functioning, fatigue, nausea and vomiting, pain, insomnia, appetite loss, constipation, and financial difficulties among the four timepoints (*p* < 0.001).

Multiple linear regression analysis (method: enter) was used in order to identify the predictors in performance status, in the follow-up period (4th assessment). Only one factor was found to be statistically significant, the grade of the cancer (Table 4). Although there is a statistically significant change in QOL over time, no other demographic or clinical factors seemed to affect any of the QOL measures. 

## 3. Discussion

This study investigated the incidence of QOL in Greek oral cancer patients before and after surgical therapy. It contributes to the growing body of evidence regarding this issue and provides important information for Greek health care professionals. 

In this study we found that the most common site of tumor was the tongue (41.5%) following by the jaw (21%). These results were similar to studies conducted by Pingili et al. [7] who found that the common sites were the tongue (35%), followed by the floor of the mouth (18.6%).

We examined the associations between demographic characteristics and subscales of QOL and we found that age correlated with emotional functioning. To the best of our knowledge, age affected the overall QOL [18] but there is not clear evidence regarding which aspect of QOL is most affected. Hence, there is a need for further research in order to draw a safe conclusion.

In addition, we found that the location of tumor affected social contact in first timepoint (pre-surgery) and nutritional supplements on the fourth assessment (12 months after surgery). This is an expected outcome because there is an argument that oral cancer patients experience pain, taste disturbances and similar symptoms that influence them socially and nutritionally [7]. 

The findings that stage correlated with fatigue and pain and also correlated with the general functioning of FACT on third measurement point (six month after surgery) were not consistent with the findings of a recent study [8]. In this study we found that stage of cancer affected QOL particularly three and six months after surgery. This discrepancy maybe explained by the different sample sizes and the different research methods. There is a need for further research in order to clarify this issue in Greek oral cancer patients. 

We found that other demographic factors such as educational status, gender and family status did not affect quality of life. This is inconsistent with the findings of other studies [18,19]. This discrepancies across the literature might be explained by the differences in the sample size or different questionnaires that were used across the studies. Moreover, real-life social differences, health perceptions and social environment support differences among different countries might have a role in these discrepancies. There is a need for further research in order to clarify the influence of gender in QOL of Greek oral cancer patients. Although it has been reported that marital environment support has an impact on the improvement of the clinical condition in cancer patients [19], it was not possible to confirm the latter argument in this study. 

It is worthwhile to note that the study group is very heterogeneous so we cannot draw safe conclusions regarding age and educational status. There is a great need for further research in order to draw safe conclusions. 

According to the trajectory of quality of life, we found that the QOL deteriorates immediately after treatment but significantly improves over time. This is in line with the findings of some recent studies [7,20] and partially with the findings of other studies [8,9]. Further studies are needed to clarify the trajectory of QOL in oral cancer patients in Greece.

In the present study, the scores of almost all of symptoms (except for coughing) were increased in the first measurement point (after 3 months) and reduced during the other measurement points. These findings are expected outcomes and maybe attributed to treatment and its adverse effects. It is worthwhile to note that all these patients underwent radiotherapy that caused a variety of adverse effects. It can be inferred that the adverse effects of treatment such as extra oral surgical scars, pigmented skin, and alopecia which can significantly impact the psychological well-being of individuals are severe immediately after treatment but improve subsequently over time.

Furthermore, in the present study we found that the financial difficulties of the patients seemed to worsen between the first and second timepoint, followed by improvement. The costs associated with the treatment of oral cancer cause financial difficulties for patients. Felder and Bennett (2013), conducting a survey on the financial difficulties of patients related to the cost of treatment, found that its cost can affect compliance, while at the same time due to the time required for treatment, patients are forced to look for alternative ways of funding [21]. Financial difficulties can be a result of many patients being required to leave their occupation in order to have their treatment

Regarding the results of the regression analysis, the factor that influenced the performance status on the 4th assessment was the grade of the tumor. Athough this result is an expected outcome, it came as a surprise to us, that stage or lymphadenectomy was not included among the predictor factors. This is inconsistent with the findings of another study [4]. There is a great need for further research in order to clarify this issue.

This study has some limitations. It was conducted in a single tertiary referral hospital located in a major city in Northern Greece, meaning that the results cannot be generalized to the entire Greek population. Another limitation is that although the study was prospective, we could not assess the impact of radiotherapy on the QOL and whether its adverse effects influence QOL in this group of Greek patients as timepoints were scheduled with reference to operation day. However, the results provide valuable information for the issue at hand and illustrate the great need for further longitudinal studies in order to draw reliable conclusions. Despite these limitations, our study has one significant strength: to our knowledge, this is the first population-based study to investigate the trajectory and pattern of quality of life, as well as the factors that influence the QOL in oral cancer patients in Greece, where the culture and lifestyle can be significantly different from those in other populations.

## 4. Conclusions

In this longitudinal study, we found that the QOL in patients treated for oral cancer deteriorates immediately after treatment but significantly improves over time. Our study also highlights the importance of the stage of cancer and the location of tumor in terms of their influence on the patients’ quality of life. Health care professionals need to be more aware of QOL issues within this group of cancer patients in order to meet their needs. They should arrange care plans and take into account these factors in order to find ways to increase these patients’ quality of life. Further research is needed to exam the trajectory of QOL in oral cancer patients in other oncology hospitals in Greece, which could add important information to the Greek oncology literature.

## Figures and Tables

**Table 1 jcm-11-07235-t001:** Demographic characteristics of patient sample.

Variables	n	%
Gender		
Male	84	62.2
Female	51	37.8
Family status		
Single	15	11.4
Married/With partner	63	47.8
Divorced	18	13.6
Widowed	36	27.3
Education		
No education (illiterate)	4	3
Primary school (6 years)	53	40.2
Middle school (9 years)	29	22
High school (12 years)	30	22.8
University (16 years)	15	11.4
Master’s/PhD	1	0.8
Occupation		
Unemployed	17	12.8
Private sector	18	13.6
Public sector	5	3.8
Self-employed	13	9.8
In retirement	79	59.8
Residence		
City	75	56.8
Town	18	13.6
Village	39	29.5

**Table 2 jcm-11-07235-t002:** Clinical characteristics of the sample.

Variables	n	%
Location		
Tongue	56	41.5
Jaw	29	21.5
Floor	16	11.9
Cheek (mucosa)	21	15.6
Lip	13	9.6
Stage		
I	11	11.8
II	21	22.6
III	12	12.9
IV	49	52.7
Grade		
1	53	43.4
2	55	45.1
3	14	11.5
Lymphadenectomy		
No	3	2.2
Levels only	105	77.8
Modified MRNDI-V	27	20.0

**Table 3 jcm-11-07235-t003:** QOL subscales in three measurement points.

	1st Timepoint (Baseline)		2nd Timepoint (3 Months)		3rd Timepoint (6 Months)		4th Timepoint (1 Year)		
	Median	Range	Median	Range	Median	Range	Median	Range	
**EORTCQLQ-30**									
Global health status	66.7	100	50.0	75.0	66.7	100.0	66.7	100.0	*χ^2^*(3) = 103.9 *p* < 0.001
Physical functioning	86.7	80	80.0	86.7	93.3	80.0	93.3	80.0	*χ^2^*(3) = 62.5 *p* < 0.001
Role functioning	66.7	100	33.3	100.0	66.7	100.0	66.7	100.0	*χ^2^*(3) = 74.4 *p* < 0.001
Emotional functioning	58.3	100	50.0	100.0	75.0	100.0	75.0	100.0	*χ^2^*(3) = 67.0 *p* < 0.001
Cognitive functioning	83.3	83.3	83.3	83.3	100.0	83.3	100.0	83.3	*χ^2^*(3) = 41.7 *p* < 0.001
Social functioning	66.7	83.3	33.3	100.0	66.7	100.0	66.7	100.0	*χ^2^*(3) = 78.1 *p* < 0.001
Fatigue	22.3	88.9	33.3	100.0	22.2	66.7	22.2	66.7	*χ^2^*(3) = 86.4 *p* < 0.001
Nausea and vomiting	0.0	50	0.0	50.0	0.0	33.3	0.0	33.3	*χ^2^*(3) = 22.0 *p* < 0.001
Pain	33.3	100	50.0	83.3	16.7	100.0	16.7	100.0	*χ^2^*(3) = 105.4 *p* < 0.001
Dyspnoea	0.0	100	33.3	66.7	0.0	100.0	0.0	100.0	*χ^2^*(3) = 6.6 *p* = 0.088
Insomnia	33.3	100	33.3	100.0	33.3	100.0	33.3	100.0	*χ^2^*(3) = 28.4 *p* < 0.001
Appetite loss	33.3	100	33.3	100.0	0.0	66.7	0.0	66.7	*χ^2^*(3) = 56.8 *p* < 0.001
Constipation	0.0	100	33.3	100.0	0.0	100.0	0.0	100.0	*χ^2^*(3) = 16.5 *p* = 0.001
Diarrhoea	0.0	100	0.0	33.3	0.0	33.3	0.0	33.3	*χ^2^*(3) = 2.5 *p* = 0.472
Financial difficulties	33.3	100	66.7	100.0	33.3	100.0	33.3	100.0	*χ^2^*(3) = 86.0 *p* < 0.001
**QLQ-HN35**									
HN-Pain	25.0	83.3	41.7	66.7	16.7	66.7	16.7	66.7	*χ^2^*(3) = 98.0 *p* < 0.001
HN-Swallowing	16.7	83.3	33.3	100.0	16.7	100.0	16.7	100.0	*χ^2^*(3) = 62.5 *p* < 0.001
HN- Sensory problems	0.0	66.7	33.3	100.0	16.7	83.3	16.7	83.3	*χ^2^*(3) = 107.7 *p* < 0.001
HN-Speech problems	0.0	66.7	22.2	55.6	11.1	77.8	11.1	77.8	*χ^2^*(3) = 63.1 *p* < 0.001
HN-Trouble with social eating	25.0	100	50.0	100.0	25.0	100.0	25.0	100.0	*χ^2^*(3) = 90.2 *p* < 0.001
HN-Trouble with social contact	0.0	60	33.3	100.0	6.7	86.7	6.7	86.7	*χ^2^*(3) = 109.8 *p* < 0.001
HN-Less sexuality	33.3	100	66.7	100.0	33.3	100.0	33.3	100.0	*χ^2^*(3) = 39.0 *p*< 0.001
HN- Teeth	0.0	100	33.3	100.0	33.3	100.0	33.3	100.0	*χ^2^*(3) = 2.7 *p* = 0.0436
HN-Trismus	0.0	100	33.3	100.0	33.3	66.7	33.3	66.7	*χ^2^*(3) = 81.5 *p* < 0.001
HN-Dry mouth	33.3	100	66.7	100.0	33.3	100.0	33.3	100.0	*χ^2^*(3) = 55.1 *p* < 0.001
HN-Sticky saliva	33.3	100	66.7	100.0	33.3	100.0	33.3	100.0	*χ^2^*(3) = 59.8 *p* < 0.001
HN-Coughing	0.0	100	0.0	66.7	0	66.7	0.0	66.7	*χ^2^*(3) = 2.8 *p* = 0.424
HN-felt ill	33.3	100	66.7	100.0	33.3	100.0	33.3	100.0	*χ^2^*(3) = 74.7 *p* < 0.001
HN-Pain Killers	33.3	33.3	0.0	33.3	0.0	33.3	0.0	33.3	*χ^2^*(3) = 65.1 *p* < 0.001
HN- Nutritional supplements	0.0	33.3	33.3	33.3	0.0	33.3	0.0	33.3	*χ^2^*(3) = 25.2 *p* < 0.001

Bonferroni correction, statistically significant: α = 0.05/39 = 0.001.

**Table 4 jcm-11-07235-t004:** Linear regression model with predictor “grade” for performance status at the 4th timepoint assessment (follow-up).

Model	B	Std. Error	t	*Sig.*
(Constant)	101.92	3.99	25.57	0.000
Grade	−6.02	2.22	−2.72	0.008

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
