# Peer review of "Quality of Life in Oral Cancer Patients in Greek Clinical Practice: A Cohort Study"

_jcm, 2022, doi:10.3390/jcm11237235_

Round 1
Reviewer 1 Report
I read the article with great interest. I feel,
Some corrections are needed -
1. Η=22,49, p=0,000 ..... everywhere should match with a dot rather than a comma.
2. QOL.... abbreviation first then use the short form.
3. Questionnaire should be added.
4. Translation validation is a must
5. Reliability and validity tests are needed.
6. Sample size - power sample size calculations should be added.
7. IRB number should be added.
9. Data are very old. Should try to update it with the latest data. 2019 and 2022
Author Response
Dear Mrs/Sir
we would like to thank you for your comments that lead to correct our project
1.Η=22,49, p=0,000 ..... everywhere should match with a dotrather than a comma. Thank you for your comment. We corrected in the text
2. QOL.... abbreviation first then use the short form. We corrected in the text
3. Questionnaire should be added. We added the full name of the questionnaires
4. Translation validation is a must. We added in the text information regard to translation and validation in Greek language
5. Reliability and validity tests are needed. In the text there is information about Cronbach a
6. Sample size - power sample size calculations should beadded. we added in the text the appropriate information
7. Data are very old. Should try to update it with the latest data.2019 and 2022
We change the data and added newer references
Reviewer 2 Report
1. This paragraph is more appropriate for discussion than introduction:
Several studies have examined the trajectory of quality of life in oral cancer patients 49 after the surgical treatment [8,9]. One study examined 96 patients with oropharyngeal 50 cancer at diagnosis, 3, 6 and 12 months using EORTC-QLQ C30. They found that quality 51 of life to be deteriorated and stage was a factor that affected the quality of life [8]. Similar 52 results have been concluded by Oskam et al (2013), who examined 26 patients with oral 53 or oropharyngeal cancer at four timepoints: pretreatment, at 6 months, 12 months and 8- 54 11 years after the treatment [9]. 55 Also, another study reported that stage was a predictive factor of quality of life in 56 cancer patients with oral or oropharyngeal cancer. They used EORTC-QLQ C30 and H&N 57 35.questionaires [10].
2. Demographic and clinical characteristics are displayed in tables 1 and 2 respectively. The mean age of our sample was 65,5 (±13.4) years, with a range from 20 to 91 years of age.
Quality of life largely depends on the type of cancer but also on age. The age range is very wide and includes a range of more than 70 years. The study group is very heterogeneous.
3. We found that other demographic factors such as educational status, gender and 214 family status did not affect quality of life.
There were 4 people in the study group in the subgroup with no education and 1 was a subgroup with a doctorate. The number of people in the subgroups in question does not allow any conclusions to be drawn.
Author Response
Dear Sir/Mrs
Thank you for comments . We corrected the text as follows:
- we changed the text and added recent data-references, so this paragraph is ommitted. We changed it according to comments of other reviewer. However, we transfer some information to the discussion.
- the study group is heteregenous and we added a comment in the discussion about this
- we added a comment for this in the discussion
Reviewer 3 Report
This manuscript compares the quality of life with several clinical and sociodemographic variables, promoting a reflection on the oral cancer impact on patients. Overall, the authors should submit the manuscript to a proofreading service. There are English language inconsistencies and typos.
Concerning the statistical analysis, the authors must change the Kolmogorov-Smirnov test (and maybe subsequent statistical test) for more reliable normality tests (see explanation in PMID: 25722854).
Finally, significant sentences in the introduction section should rewrite or relocate to the discussion field.
Author Response
Dear Sir/Mrs
Thank you for your corrections in the article. we corrected it as following
We added more reliable normality test, we added the following sentence: Kolmogorov-Smirnov and Shapiro-Wilks tests were conducted, as well as Q-Q plots were constructed to investigate normality in data.
Also, we allocate some sentences from introduction to discussion section
Round 2
Reviewer 1 Report
Some questions are not yet addressed -
2. QOL.... abbreviation first then use the short form...not yet correct
3. Questionnaire should be added. - nothing added
4. Translation validation is a must - nothing added
5. Reliability and validity tests are needed. - wrong details added. Please consult a statistician.
6. Sample size - power sample size calculations should be added. - wrong details added. Please consult a statistician.
7. IRB number should be added...nothing added
9. Data are very old. Should try to update it with the latest data. 2019 and 2022 - nothing added
Author Response
1.QOL.... abbreviation first then use the short form...not yet. correct
Added also in abstract. Thank you.
3. Questionnaire should be added. - nothing added
EORTC-QLQ C30 and HN-35 modules where used. Now provided in a supplement table.
4. Translation validation is a must - nothing adde. Instruments are already and long before validated and culturaly in Greek. Cite Nalbadian, Kyrgidis
- Reliability and validity tests are needed. - wrong details added. Please consult a tatistician.
Cronbach's α, ISS, can provide, send me dataset sav file.
- Sample size - power sample size calculations should be added. - wrong details added. Please consult a statistician.
In this cohort study of independent cases, assuming 1:1 ratio aming different out-comes, a failure rate among patients with the exposure of 0,2, and e true measured failure rate of 0,5, we will need to study 38 times 2=76 subjects and be able to reject the null hy-pothesis that the failure rates among different groups are equal with probability (power) 0,8. With regarf to paired measurements, assuming continuous variables with standard deviation 0,4 of the mean and a true difference in the mean response of matched pairs of 0,3, we will need to study 16 pairs of subjects to be able to reject the null hypothesis that this response difference is zero with probability (power) 0,8. The Type I error probability associated with these tests of this null hypothesis is 0,05.
- IRB number should be added...nothing added
Add it.
Reviewer 3 Report
Dear authors,
Thank you for taking my suggestion to improve your study into consideration. Unfortunately, some typos persist, and we have words possibly written in Greek (definitely not in English) in Table 3 and its respective legend. Please review these: "Εύρος" and "Επίπεδο στατιστικής σημαντικότητας με διόρθωση Bonferroni: α = 0,05/39 = 0,001."
Author Response
Dear Reviewer, thank you for your comments. Its coorrected in the text